# ResNets Ensemble via the Feynman-Kac Formalism to Improve Natural and Robust Accuracies

**Bao Wang**
Department of Mathematics
University of California, Los Angeles
wangbaonj@gmail.com

**Binjie Yuan**
School of Aerospace
Tsinghua University
ybj14@mail.tsinghua.edu.cn

**Zuoqiang Shi**
Department of Mathematics
Tsinghua University
zqshi@mail.tsinghua.edu.cn

**Stanley J. Osher**
Department of Mathematics
University of California, Los Angeles
sjo@math.ucla.edu

## Abstract

We unify the theory of optimal control of transport equations with the practice of training and testing of ResNets. Based on this unified viewpoint, we propose a simple yet effective ResNets ensemble algorithm to boost the accuracy of the robustly trained model on both clean and adversarial images. The proposed algorithm consists of two components: First, we modify the base ResNets by injecting a variance specified Gaussian noise to the output of each residual mapping, and it results in a special type of neural stochastic ordinary differential equation. Second, we average over the production of multiple jointly trained modified ResNets to get the final prediction. These two steps give an approximation to the Feynman-Kac formula for representing the solution of a convection-diffusion equation. For the CIFAR10 benchmark, this simple algorithm leads to a robust model with a natural accuracy of **85.62**% on clean images and a robust accuracy of **57.94**% under the 20 iterations of the IFGSM attack, which outperforms the current state-of-the-art in defending against IFGSM attack on the CIFAR10. The code is available at https://github.com/BaoWangMath/EnResNet.

## 1 Introduction

Despite the extraordinary success of deep neural nets (DNNs) in image and speech recognition [23], their vulnerability to adversarial attacks raises concerns when applying them to security-critical tasks, e.g., autonomous cars [3, 1], robotics [14], and DNN-based malware detection systems [31, 13]. Since the seminal work of Szegedy et al. [38], recent research shows that DNNs are vulnerable to many kinds of adversarial attacks including physical, poisoning, and inference attacks [9, 7, 30, 12, 17, 5, 4].

The empirical adversarial risk minimization (EARM) is one of the most successful frameworks for adversarial defense. Under the EARM framework, adversarial defense for $\ell_\infty$ norm based inference attacks can be formulated as solving the following EARM [29, 45]

$$\min_{f \in \mathcal{H}} \frac{1}{n} \sum_{i=1}^{n} \max_{\|\mathbf{x}_i' - \mathbf{x}_i\|_\infty \leq \epsilon} L(f(\mathbf{x}_i', \mathbf{w}), y_i), \tag{1}$$

where $f(\cdot, \mathbf{w})$ is a function in the hypothesis class $\mathcal{H}$, e.g., ResNet [16], parameterized by $\mathbf{w}$. Here, $\{(\mathbf{x}_i, y_i)\}_{i=1}^{n}$ are $n$ i.i.d. data-label pairs drawn from some high dimensional unknown distribution $\mathcal{D}$, $L(f(\mathbf{x}_i, \mathbf{w}), y_i)$ is the loss associated with $f$ on $(\mathbf{x}_i, y_i)$. For classification, $L$ is typically selected to be the cross-entropy. Adversarial defense for other measure based attacks can be formulated similarly.

## 1.1 Our Contribution

In this work, we unify the training and testing of ResNets with the theory of transport equations (TEs). This unified viewpoint enables us to interpret the adversarial vulnerability of ResNets as the irregularity, which will be defined later, of the TE's solution. Based on this observation, we propose a new ResNets ensemble algorithm based on the Feynman-Kac formula. In a nutshell, the proposed algorithm consists of two essential components. First, for each $l = 1, 2, \cdots, M$ with $M$ being the number of residual mappings in the ResNet, we modify the $l$-th residual mapping from $\mathbf{x}_{l+1} = \mathbf{x}_l + \mathcal{F}(\mathbf{x}_l)$ (Fig. 1 (a)) to $\mathbf{x}_{l+1} = \mathbf{x}_l + \mathcal{F}(\mathbf{x}_l) + N(0, \sigma^2 \mathbf{I})$ (Fig. 1 (b)), where $\mathbf{x}_l$ is the input, $\mathcal{F}$ is the residual mapping and $N(0, \sigma^2 \mathbf{I})$ is Gaussian noise with a specially designed variance $\sigma^2$. This step can be regarded as building a simple neural stochastic differential equation. Second, we average over multiple jointly trained, by solving the EARM Eq. (1), modified ResNets' outputs to get the final prediction (Fig. 1 (c)). This ensemble algorithm improves the base model's accuracy on both clean and adversarial data. The advantages of the proposed algorithm are summarized as follows:

- It outperforms the current state-of-the-art in defending against inference attacks.

- It improves the natural accuracy of the adversarially trained models.

- Its defense capability can be improved dynamically as the base ResNet advances.

- It is motivated from partial differential equation (PDE) theory, which introduces a new way to defend against adversarial attacks, and it is a complement to many other existing adversarial defenses.

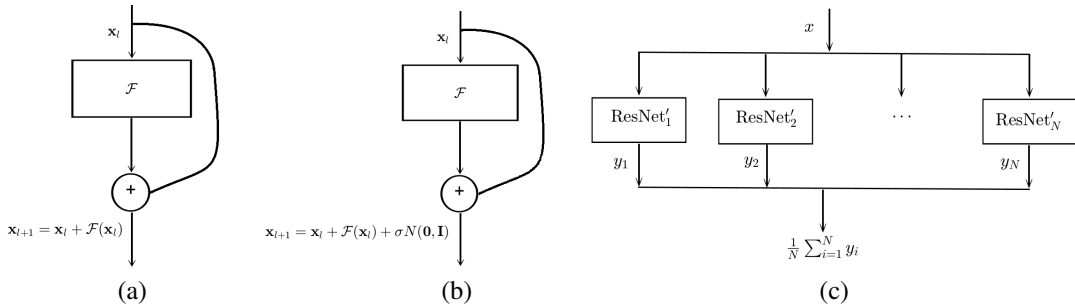

Figure 1: Original (a)/noise injected (b) residual mapping. (c) Architecture of the EnResNet.

## 1.2 Related Work

There is a massive volume of research over the last several years on defending against adversarial attacks for DNNs. Randomized smoothing transforms an arbitrary classifier $f$ into a "smoothed" surrogate classifier $g$ and is certifiably robust in $\ell_2$ norm based adversarial attacks [25, 24, 10, 43, 6, 27, 34]. One of the ideas is to inject Gaussian noise to the input image and the classification result is based on the probability of the noisy image in the decision region. Our adversarial defense algorithm injects noise into each residual mapping instead of the input image.

Robust optimization for solving EARM achieves great success in defending against inference attacks [29, 32, 33, 44, 36, 42]. Regularization in EARM can further boost the robustness of the adversarially trained models [45, 21, 35, 47]. The adversarial defense algorithms should learn a classifier with high test accuracy on both clean and adversarial data. To achieve this goal, Zhang et al. [46] developed a new loss function, TRADES, that explicitly trades off between natural and robust generalization. To the best our of knowledge, TRADES is the current state-of-the-art in defending against inference attacks on the CIFAR10. Throughout this paper, we regard TRADES as the benchmark.

Modeling DNNs as ordinary differential equations (ODEs) has drawn lots of attention recently. Chen et al. proposed neural ODEs for deep learning [8]. E [11] modeled training ResNets as solving an ODE optimal control problem. Haber and Ruthotto [15] constructed stable DNN architectures based on the properties of ODEs. Lu, Zhu and et al. [28, 48] constructed novel architectures for DNNs, which were motivated from the numerical discretization schemes for ODEs.

## 1.3 Organization

This paper is organized in the following way: In section 2, we establish the connection between training/testing of ResNets and the theory of TEs. This connection gives us a way to decipher the adversarial vulnerability of the ResNet, and we propose a simple ensemble algorithm based on the Feynman-Kac formula to enhance the guaranteed adversarial robustness of ResNet. In section 3, we test the efficiency of the proposed ResNets ensemble for adversarial defense on both CIFAR10 and CIFAR100. Section 4 contains some concluding remarks.

## 2 Algorithm and Theoretical Motivation

### 2.1 Transport Equation Modeling of ResNet

The connection between training ResNet and solving optimal control problems of the TE is investigated in [39, 40, 26, 41]. In this section, we derive the TE model for ResNet and explain its adversarial vulnerability from a PDE perspective. The TE model enables us to understand the data flow of the entire training and testing data in the forward and backward propagation; whereas, the ODE models focus on the dynamics of individual data points [8].

As shown in Fig. 1 (a), residual mapping adds a skip connection to connect the input and output of the original mapping ($\mathcal{F}$), and the $l$-th residual mapping can be written as

$$\mathbf{x}_{l+1} = \mathcal{F}(\mathbf{x}_l, \mathbf{w}_l) + \mathbf{x}_l,$$

with $\mathbf{x}_0 = \hat{\mathbf{x}} \in T \subset \mathbb{R}^d$ being a data point in the set $T$, $\mathbf{x}_l$ and $\mathbf{x}_{l+1}$ are the input and output tensors of the residual mapping. The parameters $\mathbf{w}_l$ can be learned by back-propagating the training error.

For $\forall \, \hat{\mathbf{x}} \in T$ with label $y$, the forward propagation of ResNet can be written as

$$\begin{cases} \mathbf{x}_{l+1} = \mathbf{x}_l + \mathcal{F}(\mathbf{x}_l, \mathbf{w}_l), \;\; l = 0, 1, \ldots, L-1, \;\; \text{with} \;\; \mathbf{x}_0 = \hat{\mathbf{x}}, \\ \hat{y} \doteq f(\mathbf{x}_L), \end{cases} \tag{2}$$

where $\hat{y}$ is the predicted label, $L$ is the number of layers, and $f(\mathbf{x}) = \text{softmax}(\mathbf{w}_0 \cdot \mathbf{x})$ be the output activation with $\mathbf{w}_0$ being the trainable parameters.

Next, let $t_l = l/L$, for $l = 0, 1, \cdots, L$, with interval $\Delta t = 1/L$. Without considering dimensional consistency, we regard $\mathbf{x}_l$ in Eq. (2) as the value of $\mathbf{x}(t)$ at $t_l$, so Eq. (2) can be rewritten as

$$\begin{cases} \mathbf{x}(t_{l+1}) = \mathbf{x}(t_l) + \Delta t \cdot \overline{F}(\mathbf{x}(t_l), \mathbf{w}(t_l)), \;\; l = 0, 1, \ldots, L-1, \;\; \text{with} \; \mathbf{x}(0) = \hat{\mathbf{x}} \\ \hat{y} \doteq f(\mathbf{x}(1)), \end{cases} \tag{3}$$

where $\overline{F} \doteq \frac{1}{\Delta t} \mathcal{F}$. Eq. (3) is the forward Euler discretization of the following ODE

$$\frac{d\mathbf{x}(t)}{dt} = \overline{F}(\mathbf{x}(t), \mathbf{w}(t)), \;\; \mathbf{x}(0) = \hat{\mathbf{x}}. \tag{4}$$

Let $u(\mathbf{x}, t)$ be a function that is constant along the trajectory defined by Eq. (4), then $u(\mathbf{x}, t)$ satisfies

$$\frac{d}{dt}\left(u(\mathbf{x}(t), t)\right) = \frac{\partial u}{\partial t}(\mathbf{x}, t) + \overline{F}(\mathbf{x}, \mathbf{w}(t)) \cdot \nabla u(\mathbf{x}, t) = 0, \;\; \mathbf{x} \in \mathbb{R}^d. \tag{5}$$

If we enforce the terminal condition at $t = 1$ for Eq. (5) to be

$$u(\mathbf{x}, 1) = \text{softmax}(\mathbf{w}_0 \cdot \mathbf{x}) := f(\mathbf{x}),$$

then according to the fact that $u(\mathbf{x}, t)$ is constant along the curve defined by Eq. (4) (which is called the characteristic curve for the TE defined in Eq. (5)), we have $u(\hat{\mathbf{x}}, 0) = u(\mathbf{x}(1), 1) = f(\mathbf{x}(1))$; therefore, the forward propagation of ResNet for $\hat{\mathbf{x}}$ can be modeled as computing $u(\hat{\mathbf{x}}, 0)$ along the characteristic curve of the following TE

$$\begin{cases} \frac{\partial u}{\partial t}(\mathbf{x}, t) + \overline{F}(\mathbf{x}, \mathbf{w}(t)) \cdot \nabla u(\mathbf{x}, t) = 0, \;\; \mathbf{x} \in \mathbb{R}^d, \\ u(\mathbf{x}, 1) = f(\mathbf{x}). \end{cases} \tag{6}$$

Meanwhile, the backpropagation in training ResNets can be modeled as finding the velocity field, $\overline{F}(\mathbf{x}(t), \mathbf{w}(t))$, for the following control problem

$$\begin{cases} \frac{\partial u}{\partial t}(\mathbf{x}, t) + \overline{F}(\mathbf{x}, \mathbf{w}(t)) \cdot \nabla u(\mathbf{x}, t) = 0, \;\; \mathbf{x} \in \mathbb{R}^d, \\ u(\mathbf{x}, 1) = f(\mathbf{x}), \;\; \mathbf{x} \in \mathbb{R}^d, \\ u(\mathbf{x}_i, 0) = y_i, \;\; \mathbf{x}_i \in T, \;\; \text{with } T \text{ being the training set.} \end{cases} \tag{7}$$

Note that in the above TE formulation of ResNet, $u(\mathbf{x}, 0)$ serves as the classifier and the velocity field $\overline{F}(\mathbf{x}, \mathbf{w}(t))$ encodes ResNet's architecture and weights. When $\overline{F}$ is very complex, $u(\mathbf{x}, 0)$ might be highly irregular i.e. a small change in the input $\mathbf{x}$ can lead to a massive change in the value of $u(\mathbf{x}, 0)$. This irregular function may have a good generalizability, but it is not robust to adversarial attacks. Fig. 2 (a) shows a 2D illustration of $u(\mathbf{x}, 0)$ with the terminal condition $u(\mathbf{x}, 1)$ shown in Fig. 2 (d); we will discuss this in detail later in this section.

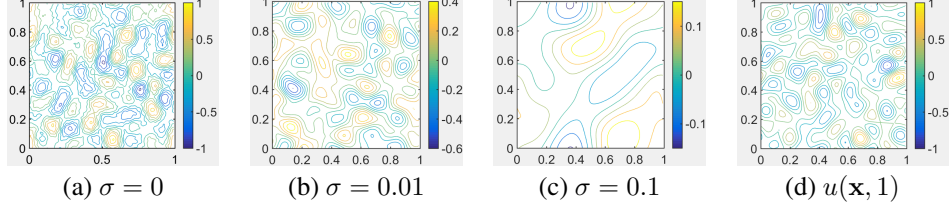

|              (a) $\sigma = 0$              |              (b) $\sigma = 0.01$              |              (c) $\sigma = 0.1$              |              (d) $u(\mathbf{x}, 1)$              |

Figure 2: (d): terminal condition for Eq. (8); (a), (b), and (c): solutions of the convection-diffusion equation, Eq. (8), at $t = 0$ with different diffusion coefficients $\sigma$.

## 2.2 Adversarial Defense by ResNets Ensemble via the Feynman-Kac Formalism

Using a specific level set of $u(\mathbf{x}, 0)$ in Fig. 2 (a) for classification suffers from adversarial vulnerability: A tiny perturbation in $\mathbf{x}$ will lead the output to go across the level set, thus leading to misclassification. To mitigate this issue, we introduce a diffusion term $\frac{1}{2}\sigma^2 \Delta u$ to Eq. (6), with $\sigma$ being the diffusion coefficient and $\Delta$ is the Laplace operator in $\mathbb{R}^d$, to make the level sets of the TE more regular. This improves robustness of the classifier. Hence, we arrive at the following convection-diffusion equation

$$\begin{cases} \frac{\partial u}{\partial t}(\mathbf{x}, t) + \overline{F}(\mathbf{x}, \mathbf{w}(t)) \cdot \nabla u(\mathbf{x}, t) + \frac{1}{2}\sigma^2 \Delta u(\mathbf{x}, t) = 0, & \mathbf{x} \in \mathbb{R}^d, \quad t \in [0, 1), \\ u(\mathbf{x}, 1) = f(\mathbf{x}). \end{cases} \tag{8}$$

The solution to Eq. (8) is much more regular when $\sigma \neq 0$ than when $\sigma = 0$. We consider the solution of Eq. (8) in a 2D unit square with periodic boundary conditions, and on each grid point of the mesh the velocity field $\overline{F}(\mathbf{x}, \mathbf{w}(t))$ is a random number sampled uniformly from $-1$ to $1$. The terminal condition is also randomly generated, as shown in Fig. 2 (d). This 2D convection-diffusion equation is solved by the pseudo-spectral method with spatial and temporal step sizes being $1/128$ and $1 \times 10^{-3}$, respectively. Figure 2 (a), (b), and (c) illustrate the solutions when $\sigma = 0$, $0.01$, and $0.1$, respectively. These show that as $\sigma$ increases, the solution becomes more regular, which makes the classifier more robust, but might be less accurate on clean data. The $\sigma$ should be selected to have a good trade-off between accuracy and robustness. Moreover, we have the following theoretical guarantee for robustness of the solution to the convection-diffusion equation.

**Theorem 1.** *[22] Let $\overline{F}(\mathbf{x}, t)$ be a Lipschitz function in both $\mathbf{x}$ and $t$, and $f(\mathbf{x})$ be a bounded function. Consider the following initial value problem of the convection-diffusion equation ($\sigma \neq 0$)*

$$\begin{cases} \frac{\partial u}{\partial t}(\mathbf{x}, t) + \overline{F}(\mathbf{x}, \mathbf{w}(t)) \cdot \nabla u(\mathbf{x}, t) + \frac{1}{2}\sigma^2 \Delta u(\mathbf{x}, t) = 0, & \mathbf{x} \in \mathbb{R}^d, \quad t \in [0, 1), \\ u(\mathbf{x}, 1) = f(\mathbf{x}). \end{cases} \tag{9}$$

*Then, for any small perturbation $\delta$, we have $|u(\mathbf{x} + \delta, 0) - u(\mathbf{x}, 0)| \leq C \left( \frac{\|\delta\|_2}{\sigma} \right)^\alpha$ for some constant $\alpha > 0$ if $\sigma \leq 1$. Here, $\|\delta\|_2$ is the $\ell_2$ norm of $\delta$, and $C$ is a constant that depends on $d$, $\|f\|_\infty$, and $\|\overline{F}\|_{L^\infty_{\mathbf{x}, t}}$.*

According to the above observation, instead of using $u(\mathbf{x}, 0)$ of the TE's solution for classification, we use that of the convection-diffusion equation. The above convection-diffusion equation can be solved using the Feynman-Kac formula [18] in high dimensional space, which gives $u(\hat{\mathbf{x}}, 0)$ as [1]

$$u(\hat{\mathbf{x}}, 0) = \mathbb{E}\left[f(\mathbf{x}(1))|\mathbf{x}(0) = \hat{\mathbf{x}}\right], \tag{10}$$

where $\mathbf{x}(t)$ is an Itô process,

$$d\mathbf{x}(t) = \overline{F}(\mathbf{x}(t), \mathbf{w}(t))dt + \sigma dB_t,$$

and $u(\hat{\mathbf{x}}, 0)$ is the conditional expectation of $f(\mathbf{x}(1))$.

We approximate the Feynman-Kac formula by an ensemble of modified ResNets in the following way: Accoding to the Euler–Maruyama method [2], the term $\sigma dB_t$ can be approximated by adding Gaussian noise $\sigma \mathcal{N}(\mathbf{0}, \mathbf{I})$, where $\sigma = a\sqrt{\text{Var}(\mathbf{x}_l + \mathcal{F}(\mathbf{x}_l))}$ with $a$ being a tunable parameter, to each residual mapping $\mathbf{x}_{l+1} = \mathbf{x}_l + \mathcal{F}(\mathbf{x}_l)$. This gives the modified residual mapping $\mathbf{x}_{l+1} = \mathbf{x}_l + \mathcal{F}(\mathbf{x}_l) + \sigma \mathcal{N}(\mathbf{0}, \mathbf{I})$, as illustrated in Fig. 1 (b). Let ResNet' denote the modified ResNet where we inject noise to each residual mapping of the original ResNet. In a nutshell, ResNet's approximation to the Feynman-Kac formula is an ensemble of jointly trained ResNet' as illustrated in Fig. 1 (c). [2] We call this ensemble of ResNets as EnResNet. For instance, an ensemble of $n$ ResNet20 is denoted as $\text{En}_n\text{ResNet20}$.

## 2.3 Robust Training of the EnResNet

We use the PGD adversarial training [29] to robustly train EnResNets with $\sigma = 0.1$ on both CIFAR10 and CIFAR100 [20] with standard data augmentation [16]. The attack in the PGD adversarial training is merely iterative fast gradient sign method (IFGSM) with an initial random perturbation on the clean data. Other methods to solve EARM can also be used to train EnResNets. All computations are carried out on a machine with a single Nvidia Titan Xp graphics card.

## 2.4 Attack Methods

We attack the trained model, $f(\mathbf{x}, \mathbf{w})$, by $\ell_\infty$ norm based untargeted FGSM, IFGSM [12], and C&W [7] attacks in both white-box and blind fashions. In blind attacks, we use the target model to classify the adversarial images crafted by attacking the oracle model in a white-box approach. For a given instance $(\mathbf{x}, y)$:

- FGSM searches the adversarial image $\mathbf{x}'$ by maximizing the loss function $\mathcal{L}(\mathbf{x}', y) \doteq \mathcal{L}(f(\mathbf{x}', \mathbf{w}), y)$, subject to the constraint $||\mathbf{x}' - \mathbf{x}||_\infty \leq \epsilon$ with $\epsilon$ being the maximum perturbation. For the linearized loss function, $\mathcal{L}(\mathbf{x}', y) \approx \mathcal{L}(\mathbf{x}, y) + \nabla_x\mathcal{L}(\mathbf{x}, y)^T \cdot (\mathbf{x}' - \mathbf{x})$, the optimal adversarial is

$$\mathbf{x}' = \mathbf{x} + \epsilon \cdot \text{sign}\left(\nabla_\mathbf{x}\mathcal{L}(\mathbf{x}, y)\right). \tag{11}$$

- IFGSM, Eq. (12), iterates FGSM with step size $\alpha$ and clips the perturbed image to generate the enhanced adversarial attack, with $\mathbf{x}^{(0)}$ being the clean data,

$$\mathbf{x}^{(m)} = \text{Clip}_{\mathbf{x}, \epsilon}\{\mathbf{x}^{(m-1)} + \alpha \cdot \text{sign}(\nabla_\mathbf{x}\mathcal{L}(\mathbf{x}^{(m-1)}, y))\}. \tag{12}$$

- C&W attack searches the targeted adversarial image by solving

$$\min_\delta ||\delta||_\infty, \quad \text{subject to} \ f(\mathbf{w}, \mathbf{x} + \delta) = t, \ \mathbf{x} + \delta \in [0, 1]^d, \tag{13}$$

where $\delta$ is the adversarial perturbation and $t$ is the target label. Carlini et al. [7] proposed the following approximation to Eq. (13),

$$\min_\mathbf{u} ||\frac{1}{2}(\tanh(\mathbf{u}) + 1) - \mathbf{x}||_\infty + c \cdot \max\left\{-\kappa, \max_{i \neq t}(Z(\frac{1}{2}(\tanh(\mathbf{u})) + 1)_i) - Z(\frac{1}{2}(\tanh(\mathbf{u})) + 1)_t\right\}, \tag{14}$$

where $Z(\cdot)$ is the logit vector for the input, i.e., the output of the DNN before the softmax layer. This unconstrained optimization can be solved efficiently by using the Adam optimizer [19].

In the following experiments, we set $\epsilon = 8/255$ in both FGSM and IFGSM attacks. Additionally, in IFGSM we set $m = 20$ and $\alpha = 2/255$, and denote it as $\text{IFGSM}^{20}$. For C&W attack, we run 50 iterations of Adam with learning rate $6 \times 10^{-4}$ and set $c = 10$ and $\kappa = 0$.

## 3 Numerical Results

In this section, we numerically verify that the robustly trained EnResNets are more accurate, on both clean and adversarial data, than robustly trained ResNets and ensemble of ResNets without noise

injection [3]. To avoid the gradient mask issue of EnResNets due to the injected noise, we use the Expectation over Transformation (EOT) strategy [5] to compute the gradient which is averaged over five independent runs.

## 3.1 EnResNets

In robust training, we run 200 epochs of the PGD adversarial training (10 iterations of IFGSM with $\alpha = 2/255$ and $\epsilon = 8/255$, and an initial random perturbation of magnitude $\epsilon$) with initial learning rate $0.1$, which decays by a factor of $10$ at the 80th, 120th, and 160th epochs. The training data is split into 45K/5K for training and validation, the model with the best validation accuracy is used for testing. $En_1ResNet20$ denotes the ensemble of only one ResNet20 which is merely adding noise to each residual mapping, and similar notations apply to other DNNs.

First, consider natural ($\mathcal{A}_{\text{nat}}$) and robust ($\mathcal{A}_{\text{rob}}$) accuracies of the PGD adversarially trained models on the CIFAR10, where $\mathcal{A}_{\text{nat}}$ and $\mathcal{A}_{\text{rob}}$ are measured on clean and adversarial images, respectively. All results are listed in Table 1. The robustly trained ResNet20 has accuracies $50.89\%$, $46.03\%$ (close to that reported in [29]), and $58.73\%$, respectively, under the FGSM, IFGSM[20], and C&W attacks. Moreover, it has a natural accuracy of $75.11\%$. $En_5ResNet20$ boosts natural accuracy to $82.52\%$, and improves the corresponding robust accuracies to $58.92\%$, $51.48\%$, and $67.73\%$, respectively. Simply injecting noise to each residual mapping of ResNet20 can increase $\mathcal{A}_{\text{nat}}$ by $\sim 2\%$ and $\mathcal{A}_{\text{rob}}$ by $\sim 3\%$ under the IFGSM[20] attack. The advantages of EnResNets are also verified by experiments on ResNet44, ResNet110, and their ensembles. Note that ensemble of high capacity ResNet is more robust than low capacity model: as shown in Table 1, $En_2ResNet110$ is more accurate than $En_2ResNet44$ which in turn is more accurate than $En_2ResNet20$ in classifying both clean and adversarial images. The robustly trained $En_1WideResNet34-10$ has $86.19\%$ and $56.60\%$, respectively, natural and robust accuracies under the IFGSM[20] attack. Compared with the current state-of-the-art [46], $En_1WideResNet34-10$ has almost the same robust accuracy ($56.60\%$ v.s. $56.61\%$) under the IFGSM[20] attack but better natural accuracy ($86.19\%$ v.s. $84.92\%$). Figure 3 plots the evolution of training and validation accuracies of ResNet20 and ResNet44 and their different ensembles.

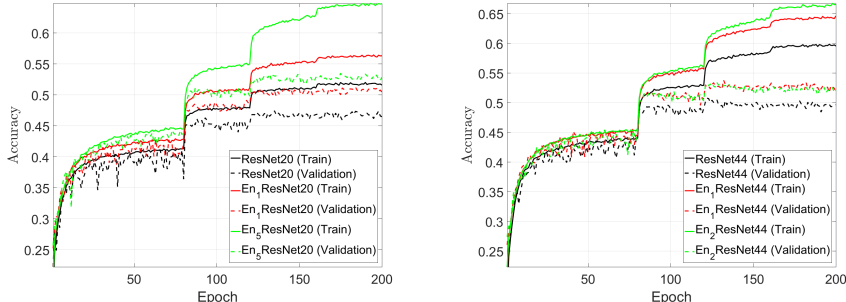

Figure 3: Evolution of training and validation accuracies. Left: ResNet20 and different ensembles of noise injected ResNet20. Right: ResNet44 and different ensembles of noise injected ResNet44.

Second, consider the robustly trained models under blind attacks. In this scenario, we use the target model to classify the adversarial images crafted by applying FGSM, IFGSM[20], and C&W attacks to the oracle model. As listed in Table 2, EnResNets are always more robust than the base ResNets under different blind attacks. For instance, when $En_5ResNet20$ is used to classify adversarial images crafted by attacking ResNet20 with FGSM, IFGSM[20], and C&W attacks, the accuracies are $64.07\%$, $62.99\%$, and $76.57\%$, respectively. Conversely, the accuracies of ResNet20 are only $61.69\%$, $58.74\%$, and $73.77\%$, respectively.

Third, we perform experiments on the CIFAR100 to further verify the efficiency of EnResNets in defending against adversarial attacks. Table 3 lists natural and robust accuracies of ResNet20, ResNet44, and their ensembles under white-box attacks. The robust accuracy under the blind attacks is listed in Table 4. The natural accuracy of the PGD adversarially trained baseline ResNet20 is $46.02\%$, and it has robust accuracies $24.77\%$, $23.23\%$, and $32.42\%$ under FGSM, IFGSM[20], and

Table 1: Natural and robust accuracies of different base and noise injected ensembles of robustly trained ResNets on the CIFAR10. Unit: %.

| Model | dataset | $\mathcal{A}_{\text{nat}}$ | $\mathcal{A}_{\text{rob}}$ (FGSM) | $\mathcal{A}_{\text{rob}}$ (IFGSM$^{20}$) | $\mathcal{A}_{\text{rob}}$ (C&W) |
|---|---|---|---|---|---|
| ResNet20 | CIFAR10 | 75.11 | 50.89 | 46.03 | 58.73 |
| En$_1$ResNet20 | CIFAR10 | 77.21 | 55.35 | 49.06 | 65.69 |
| En$_2$ResNet20 | CIFAR10 | 80.34 | 57.23 | 50.06 | 66.47 |
| En$_5$ResNet20 | CIFAR10 | 82.52 | 58.92 | 51.48 | 67.73 |
| ResNet44 | CIFAR10 | 78.89 | 54.54 | 48.85 | 61.33 |
| En$_1$ResNet44 | CIFAR10 | 82.03 | 57.80 | 51.83 | 66.00 |
| En$_2$ResNet44 | CIFAR10 | 82.91 | 58.29 | 51.86 | 66.89 |
| ResNet110 | CIFAR10 | 82.19 | 57.61 | 52.02 | 62.92 |
| En$_2$ResNet110 | CIFAR10 | 82.43 | 59.24 | 53.03 | 68.67 |
| En$_1$WideResNet34-10 | CIFAR10 | **86.19** | **61.82** | **56.60** | **69.32** |

Table 2: Accuracies of robustly trained models on adversarial images of CIFAR10 crafted by attacking the oracle model with different attacks. Unit: %.

| Model | dataset | Oracle | $\mathcal{A}_{\text{rob}}$ (FGSM) | $\mathcal{A}_{\text{rob}}$ (IFGSM$^{20}$) | $\mathcal{A}_{\text{rob}}$ (C&W) |
|---|---|---|---|---|---|
| ResNet20 | CIFAR10 | En$_5$ResNet20 | 61.69 | 58.74 | 73.77 |
| En$_5$ResNet20 | CIFAR10 | ResNet20 | 64.07 | 62.99 | 76.57 |
| ResNet44 | CIFAR10 | En$_2$ResNet44 | 63.87 | 60.66 | 75.83 |
| En$_2$ResNet44 | CIFAR10 | ResNet44 | 64.52 | 61.23 | 76.99 |
| ResNet110 | CIFAR10 | En$_2$ResNet110 | 64.19 | 61.80 | 75.19 |
| En$_2$ResNet110 | CIFAR10 | ResNet110 | 66.26 | 62.89 | 77.71 |

C&W attacks, respectively. En$_5$ResNet20 increases them to $51.72\%$, $31.64\%$, $27.80\%$, and $40.44\%$, respectively. The ensemble of ResNets is more effective in defending against adversarial attacks than making the ResNets deeper. For instance, En$_2$ResNet20 that has $\sim 0.27M \times 2$ parameters is much more robust to adversarial attacks, FGSM ($30.20\%$ v.s. $28.40\%$), IFGSM$^{20}$ ($26.25\%$ v.s. $25.81\%$), and C&W ($40.06\%$ v.s. $36.06\%$), than ResNet44 with $\sim 0.66M$ parameters. Under blind attacks, En$_2$ResNet20 is also significantly more robust to different attacks where the opponent model is used to generate adversarial images. Under the same model and computation complexity, EnResNets is more robust to adversarials and more accurate on clean images than deeper nets.

Table 3: Natural and robust accuracies of robustly trained ResNet20 and different ensemble of noise injected ResNet20 on the CIFAR100. Unit: %.

| Model | dataset | $\mathcal{A}_{\text{nat}}$ | $\mathcal{A}_{\text{rob}}$ (FGSM) | $\mathcal{A}_{\text{rob}}$ (IFGSM$^{20}$) | $\mathcal{A}_{\text{rob}}$ (C&W) |
|---|---|---|---|---|---|
| ResNet20 | CIFAR100 | 46.02 | 24.77 | 23.23 | 32.42 |
| En$_2$ResNet20 | CIFAR100 | 50.68 | 30.20 | 26.25 | 40.06 |
| En$_5$ResNet20 | CIFAR100 | **51.72** | **31.64** | **27.80** | **40.44** |
| ResNet44 | CIFAR100 | 50.38 | 28.40 | 25.81 | 36.06 |

Table 4: Accuracies of robustly trained models on the adversarial images of CIFAR100 crafted by attacking the oracle model with different attacks. Unit: %.

| Model | dataset | Oracle | $\mathcal{A}_{\text{rob}}$ (FGSM) | $\mathcal{A}_{\text{rob}}$ (IFGSM$^{20}$) | $\mathcal{A}_{\text{rob}}$ (C&W) |
|---|---|---|---|---|---|
| ResNet20 | CIFAR100 | En$_2$ResNet20 | 33.08 | 30.79 | 41.52 |
| En$_2$ResNet20 | CIFAR100 | ResNet20 | 34.15 | 33.34 | 48.21 |

Figure 4 depicts a few selected images from the CIFAR10 and their adversarial ones crafted by applying either IFGSM$^{20}$ or C&W attack to attack both ResNet20 and En$_5$ResNet20. Both adversarially trained ResNet20 and En$_5$ResNet20 fail to correctly classify any of the adversarial versions of these four images. For the deer image, both ResNet and En$_5$ResNet have only slightly higher confidence to classify them as a deer than as a horse, and it might also be difficult for a human to distinguish it from a horse.

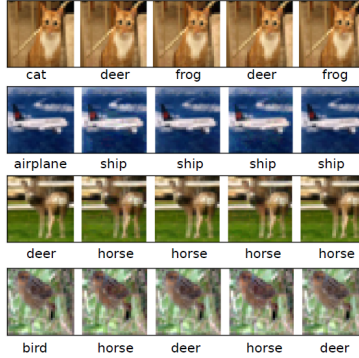

Figure 4: Column 1: original images and labels; column 2-3 (4-5): adversarial images crafted by using IFGSM[20] and C&W to attack ResNet20 (En$_5$ResNet20) and corresponding predicted labels.

## 3.2 Integration of Separately Trained EnResNets

In the previous subsection, we verified the adversarial defense capability of EnResNet, which is an approximation to the Feynman-Kac formula to solve the convection-diffusion equation. As we showed, when more ResNets and larger models are involved in the ensemble, both natural and robust accuracies are improved. However, EnResNet proposed above requires to train the ensemble jointly, which poses memory challenges for training ultra-large ensembles. To overcome this issue, we consider training each component of the ensemble individually and integrating them together for prediction. The major benefit of this strategy is that with the same amount of GPU memory, we can train a much larger model for inference since the batch size used in inference can be one.

Table 5 lists natural and robust accuracies of the integration of separately trained EnResNets on the CIFAR10. The integration have better robust accuracy than each component. For instance, the integration of En$_2$ResNet110 and En$_1$WideResNet34-10 gives a robust accuracy **57.94%** under the IFGSM[20] attack, which is remarkably better than both En$_2$ResNet110 (53.05%) and En$_1$WideResNet34-10 (56.60%). To the best of our knowledge, 57.94% outperforms the current state-of-the-art [46] by 1.33%. The effectiveness of the integration of separately trained EnResNets sheds light on the development of ultra-large models to improve efficiency for adversarial defense.

Table 5: Natural and robust accuracies of different integration of different robustly trained EnResNets on the CIFAR10. Unit: %.

| Model | dataset | $\mathcal{A}_{nat}$ | $\mathcal{A}_{rob}$ (FGSM) | $\mathcal{A}_{rob}$ (IFGSM[20]) | $\mathcal{A}_{rob}$ (C&W) |
|---|---|---|---|---|---|
| En$_2$ResNet20&En$_5$ResNet20 | CIFAR10 | 82.82 | 59.14 | 53.15 | 68.00 |
| En$_2$ResNet44&En$_5$ResNet20 | CIFAR10 | 82.99 | 59.64 | 53.86 | 69.36 |
| En$_2$ResNet110&En$_5$ResNet20 | CIFAR10 | 83.57 | 60.63 | 54.87 | 70.02 |
| En$_2$ResNet110&En$_1$WideResNet34-10 | CIFAR10 | **85.62** | **62.48** | **57.94** | **70.20** |

## 3.3 Gradient Mask and Comparison with Simple Ensembles

Besides applying EOT gradient, we further verify that our defense is not due to obfuscated gradient. We use IFGSM[20] to attack naturally trained (using the same approach as that used in [16]) En$_1$ResNet20, En$_2$ResNet20, and En$_5$ResNet20, and the corresponding accuracies are: 0%, 0.02%, and 0.03%, respectively. All naturally trained EnResNets are easily fooled by IFGSM[20], thus gradient mask does not play an important role in EnResNets for adversarial defense [4]. However, under the FGSM attack with $\epsilon = 8/255$, the naturally trained En$_1$ResNet20 and En$_2$ResNet20 (with injected Gaussian noise of standard deviation 0.1) has robust accuracies 27.93% and 28.75%, resp., and it is significantly higher than that of the naturally trained ResNet20. These results show that the naturally trained EnResNets are also more resistant to adversarial attacks.

Ensemble of models for adversarial defense has been studied in [37]. Here, we show that ensembles of robustly trained ResNets without noise injection cannot boost natural and robust accuracy much. The natural accuracy of jointly (separately) adversarially trained ensemble of two ResNet20 without noise injection is 75.75% (74.96%), which does not substantially outperform ResNet20 with a natural

accuracy 75.11%. The corresponding robust accuracies are 51.11% (51.68%), 47.28% (47.86%), and 59.73% (59.80%), respectively, under the FGSM, IFGSM$^{20}$, and C&W attacks. These robust accuracies are much inferior to that of En$_2$ResNet20. Furthermore, the ensemble of separately trained robust ResNet20 and robust ResNet44 gives a natural accuracy of 77.92%, and robust accuracies are 54.73%, 51.47%, 61.77% under the above three attacks. These results reveal that ensemble adversarially trained ResNets via the Feynman-Kac formalism is much more accurate than standard ensemble in both natural and robust generalizations.

### 3.4 Memory and Inference Time Consumption

EnResNets has negligible overhead in inference time and memory compared with inference by the standard ResNet. For instance, the inference time of ResNet20 and En$_1$ResNet20 for CIFAR10 classification, averaged over 100 runs, with batch size 1K on a Titan Xp are 1.6941s and 1.6943s, resp. The corresponding peak memory is 4807MB for both ResNet20 and En$_1$ResNet20.

### 3.5 Sensitivity to the Noise Injection

Now, we consider the effects of the injected Gaussian noise, with standard deviation $\sigma$, Table 6 lists $\mathcal{A}_{\mathrm{rob}}$ (IFGSM$^{20}$) of the robustly trained En$_2$ResNet20 with different $\sigma$. 0.1 gives a good trade-off between accuracy and variance.

Table 6: Robust accuracy of the PGD adversarially trained En$_2$ResNet20 with different Gaussian noise injection. (five runs)

| $\sigma$ | 0.05 | 0.1 | 0.4 | 0.8 |
|---|---|---|---|---|
| $\mathcal{A}_{\mathrm{rob}}$ (IFGSM$^{20}$) | 50.05% ± 0.27% | 50.06% ± 0.35% | 50.51% ± 0.90% | 43.51% ± 3.78% |

### 3.6 Sharing Weights Ensemble

Finally, we need to point out that the direct ResNet ensemble counterpart of the Feynman-Kac formalism needs to share weights. Table 7 shows that, the share weights ensemble (SWE) also improves both natural and robust accuracies which verifies the efficacy of our PDE formalism. Moreover, to further improve the ensemble model's performance, we generalize SWE to non-share weights ensemble (NSWE) with the consideration of increasing the model capacity.

Table 7: Accuracy of the robustly trained $n\times$En$_1$ResNet20 which denotes the ensemble of $n$ share-weights En$_1$ResNet20.

| | ResNet20 | $1\times$ En$_1$ResNet20 | $2\times$ En$_1$ResNet20 | $5\times$ En$_1$ResNet20 |
|---|---|---|---|---|
| $\mathcal{A}_{\mathrm{nat}}$ | 75.11% | 77.21% | 77.88% | 77.99% |
| $\mathcal{A}_{\mathrm{rob}}$ (IFGSM$^{20}$) | 46.03% | 49.06% | 49.17% | 49.20% |

## 4 Conclusions

In this paper, we utilize a transport equation to model ResNets' data flow. The lack of regularity of the transport equation's solution explains ResNets' adversarial vulnerability. The analogy of regularizing the solution of transport equation by adding a diffusion term motivates us to propose a ResNets ensemble based on the Feynman-Kac formula. The adversarially trained EnResNet remarkably improves both natural and robust accuracies towards adversarial attacks. Our method is a complement to many existing adversarial defense algorithms, for instance, directly replacing the cross-entropy loss with the TRADES loss [46] can further improve the robust accuracy, under the IFGSM$^{20}$ attack, of the WideResNet used above by $\sim 0.9\%$. As a future work, we propose to combine EnResNet with the surrogate loss function design and regularization [45].

## Acknowledgments

Bao Wang thanks Dr. Jiajun Tong and Dr. Yuming Zhang for stimulating discussion on the stability theorem for the convection-diffusion equation.

## Footnotes

[1]A detailed derivation is available in the supplementary material.

[2] To ease the notation, in what follows, we use ResNet in place of ResNet' when there is no ambiguity.

[3]The baseline ResNet implementation is available at `https://github.com/akamaster/pytorch_resnet_cifar10/blob/master/resnet.py`

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
