[Supplementary Material · EnResNet_NIPS2019_Supplementary.pdf]

# Supplementary material:
# *ResNets Ensemble via the Feynman-Kac Formalism to Improve Natural and Robust Accuracies*

**Bao Wang**
Department of Mathematics
University of California, Los Angeles
wangbaonj@gmail.com

**Binjie Yuan**
School of Aerospace
Tsinghua University
ybj14@mail.tsinghua.edu.cn

**Zuoqiang Shi**
Department of Mathematics
Tsinghua University
zqshi@mail.tsinghua.edu.cn

**Stanley J. Osher**
Department of Mathematics
University of California, Los Angeles
sjo@math.ucla.edu

## 1 Feynman-Kac Formula Representation of the Convection-Diffusion Equation's Solution

In this section, we will give a detailed discussion of using the Feynman-Kac formula to present the solution of the convection-diffusion equation.

**Lemma 1** (Feynman-Kac formula). *([1]) Consider the partial differential equation*

$$\frac{\partial u}{\partial t}(\mathbf{x}, t) + \mu(x, t)\frac{\partial u}{\partial \mathbf{x}}(\mathbf{x}, t) + \frac{1}{2}\sigma^2(\mathbf{x}, t)\frac{\partial^2 u}{\partial \mathbf{x}^2}(\mathbf{x}, t) - V(\mathbf{x}, t)u(\mathbf{x}, t) + f(\mathbf{x}, t) = 0,$$

*defined for all $\mathbf{x} \in \mathbb{R}^d$ and $t \in [0, T]$, subject to the terminal condition*

$$u(\mathbf{x}, T) = \psi(\mathbf{x}),$$

*where $\mu, \sigma, \psi, V, f$ are known functions, $T$ is a parameter and $u : \mathbb{R}^d \times [0, T] \to \mathbb{R}$ is the unknown. Then the Feynman-Kac formula tells us that the solution can be written as a conditional expectation*

$$u(\mathbf{x}, t) = \mathbb{E}\left[\int_t^T e^{-\int_t^\tau V(X_\tau, \tau)d\tau} f(X_r, r)dr + e^{-\int_t^T V(X_\tau, \tau)d\tau}\psi(X_T)|X_t = \mathbf{x}\right] \qquad (1)$$

*where $X(t)$ is an Itô process,*

$$dX = \mu(X, t)dt + \sigma(X, t)dB_t,$$

*with $B_t$ is a Wiener process, or Brownian motion.*

Now, we consider using the Feynman-Kac formula to represent the solution of the following convection-diffusion equation which is used to model the ResNet

$$\begin{cases} \frac{\partial u}{\partial t}(\mathbf{x}, t) + \overline{F}(\mathbf{x}, \mathbf{w}(t)) \cdot \nabla u(\mathbf{x}, t) + \frac{1}{2}\sigma^2 \Delta u(\mathbf{x}, t) = 0, & \mathbf{x} \in \mathbb{R}^d, \quad t \in [0, 1), \\ u(\mathbf{x}, 1) = f(\mathbf{x}). \end{cases} \qquad (2)$$

Let $t = 0$, $V(\mathbf{x}, t) = 0$, $f(\mathbf{x}, t) = 0$, and $\psi(\mathbf{x}) = f(\mathbf{x})$, we get the solution of Eq. (2) at $t = 0$, represented by the Feynman-Kac formula, as

$$u(\hat{\mathbf{x}}, 0) = \mathbb{E}\left[f(\mathbf{x}(1))|\mathbf{x}(0) = \hat{\mathbf{x}}\right], \qquad (3)$$

where $\mathbf{x}(t)$ is an Itô process,

$$d\mathbf{x}(t) = \overline{F}(\mathbf{x}(t), \mathbf{w}(t))dt + \sigma dB_t,$$

and $u(\hat{\mathbf{x}}, 0)$ is the conditional expectation of $f(\mathbf{x}(1))$.

# References

[1] M. Kac. On distributions of certain Wiener functionals. *Transactions of the American Mathematical Society*, 65:1–13, 1949.