[Reviews · NeurIPS 2019]

Reviewer 1



- Originality: This work is a novel combination of existing methods [2,18,22]. - Quality: The submission is technically sound and well supported by theoretical analysis and experimental results. Authors show some failure case examples, however, they did not particularly describe or analyze why it happened. - Clarity: The paper is easy to read. - Significance: The experimental results are important as it advances the state-of-the-art in this field. However, this method is inherently limited to ResNet based network architectures. ----------After rebuttal----------- Authors address the some concerns in rebuttal, e.g., runtime overhead is negligible. I look forward to see other results that authors promised to report in the finial version. So, I stand with my initial decision as an accept.

Reviewer 2



[Originality] Although the ideas of 1) using ensemble for adversarial robustness, 2) explaining ResNets using ODE/PDE and 3) adding noise to architectures [1,2], ares not new, to my knowledge this is the first paper that connects PDE-based theory with adversarial robustness. [Quality] The theoretical part of this paper is sound, and mostly self-contained. However, I believe there are some mismatches between the theory and the experiments. I have listed my concerns in the improvement/question section. The authors are probably not aware of some similar papers like [1,2], which possibly have great overlap with the algorithm presented by the authors. [Clarity] The theoretical part of this paper is clear. The paper provides a nice visualization for explaining how the introduced diffusion term works. Since the algorithm is rather simple, I believe one experienced reader should be able to implement that quite easily. However, I found that there may be some inconsistency between the theory and the authors' implementation: the PDE equation in Theorem 1 definitely shows that all ResNets in one single ensemble share weights, while the authors do not point this out directly. It may cause some confusion, and I will return to this point in the questions. And there are some other minor issues, like the authors do not point out the configurations for the architectures such as channel size and batch norm position. Although ResNets are quite standard, I believe it is still important to specify the details at least in the appendix. The experiment results are not easy to read. I personally believe that Section 3.3 in the paper is the worst part - the authors could have put all the numbers into a table (or into other existing ones) so that readers do not have to check back and forth. [Significance] The PDE-based approach builds a connection between recent studies on PDE/ODE for ResNets and the study of adversarial robustness. It provides some theoretical soundness for a rather simple algorithm. The algorithm itself does not seem too novel, as using ensemble and adding noise are separated presented by some other papers. I believe the best point of this paper is that it potentially shows a new direction/framework for improving adversarial robustness. Finally, there are some issues with the empirical experiments. If these issues can be answered/solved, I believe the paper can be a contribution to Neurips. [1] Liu, X., Cheng, M., Zhang, H. and Hsieh, C.J., 2018. Towards robust neural networks via random self-ensemble. In Proceedings of the European Conference on Computer Vision (ECCV) [2] He, Z., Rakin, A.S. and Fan, D., 2019. Parametric noise injection: Trainable randomness to improve deep neural network robustness against adversarial attack. In Proceedings of the IEEE Conference on Computer Vision and Pattern Recognition (pp. 588-597). [3] Cohen, J.M., Rosenfeld, E. and Kolter, J.Z., 2019. Certified adversarial robustness via randomized smoothing. arXiv preprint arXiv:1902.02918. --------Post Rebuttal---------- The authors addressed my concerns and I raised my score accordingly.

Reviewer 3



Update: I have read the author's rebuttal and the other reviews. I am happy with my previous recommendation. - Regarding experiments, the highest vanilla resnet accuracy in the paper is close to 85%, however, there are many implementations of resnets which achieve close to 95% on CIFAR10. Why aren't the baseline numbers reported in the paper high enough? The observations of the paper still hold. However, this reviewer is concerned regarding the efficacy of the method towards very high or SOTA performing models. - I believe the derivation of the solution eq 10 is important and relevant for readers new to ODEs. The paper's accessibility/correctness would be improved if the derivation was included in the supplementary. - Line 126, strong suggestion to change the symbol denoting injected noise into resnets as Resnet'. Better symbols might be Resnet* etc. This is because Line 130, uses Resnet's, which cause confusion. - How is a picked, what happens if you increase or decrease a? Minor comments: - Th 1, line 116, please add in the meanings of the symbols that were mentioned in [22]. This interrupts readability. Even a very brief one linear suffices.

[Author Response · NeurIPS 2019]

We appreciate the reviewers' support for the novelty, remarkable performance, and the impact of our work. We thank the
reviewers for their valuable feedback and thoughtful reviews. Below we address the concerns raised by the reviewers.
**Reviewer 1:** It would be great to compare inference time and memory consumption against other defense methods.
**Reply:** Our defense method has negligible overhead in inference time and memory compared with inference by the
standard ResNet without defense, and this is also the case for the PGD adversarial training (arXiv:1706.06083) and
TRADES (arXiv:1901.08573). For CIFAR10, the inference time for ResNet20 and $En_1$ResNet20, averaged over 100
runs, with batch size 1K on a Titan Xp are 1.6941s and 1.6943s, resp. The corresponding peak memory is 4807MB for
both ResNet20 and $En_1$ResNet20. We will report the inference time and memory in the revised manuscript.
**Reviewer 1:** In Fig. 4, it is unclear why neural networks made wrong predictions. It could be due to the difficulty of
the input example or adversarial attack. To make it more clear, authors need to add prediction results for the clean input.
**Reply:** We will add the classification results for the clean input in the revised manuscript.
**Reviewer 1:** I do not see experiments that can support claim that this method is a complement to existing defenses.
**Reply:** Directly replacing the cross-entropy loss with the TRADES loss can further improve the robust accuracy, under
the IFGSM[20] attack, of the WideResNet (arXiv:1901.08573) by $\sim 0.9\%$. We will report more results in the revision.
**Reviewer 1:** The method is inherently limited to ResNet based network architectures.
**Reply:** The EnResNet is motivated from the Euler-Maruyama discretization of the Itô process below Eq. (10). Other
numerical discretization may motivate ensemble of new network architectures like neural ODE as our future work.
**Reviewer 3:** The authors are probably not aware of some similar papers like [1,2,3].
**Reply:** We will discuss these papers in the related work Sec. in the revision.
**Reviewer 3:** The proposed PDE formalism and the resulted method does not depend on the training objective function.
Therefore, Theorem 1 should provide some degree of adversarial robustness even in the absence of adversarial training.
Still, empirical results regarding the performance of the proposed method with natural training are missing.
**Reply:** For natural training, $En_1$ResNet20 and $En_2$ResNet20 (with injected Gaussian noise of standard deviation 0.1)
has accuracy 27.93% and 28.75%, resp., under the FGSM attack with $\epsilon = 8/255$, in contrast to ResNet20 with robust
accuracy 10.45% under the same attack. Moreover, as shown in Sec. 3.3, the naturally trained $En_n$ResNet20 has slightly
better robust accuracy than ResNet20 under the IFGSM[20] attack. Adversarial training enables $En_n$ResNets to have
remarkably better robust accuracy than ResNets under the IFGSM[20] attack. We will make this clear in the revision.
**Reviewer 3:** I wonder if the ResNets are in a single ensemble sharing weights. If not, the implementation is not
consistent with the proposed PDE formalism. If yes, I would like to make sure that the share-weights ensemble (SWE)
of ResNets truly outperforms standard ResNet on natural examples.
**Reply:** In our paper, EnResNets do not share weights. Direct ResNet ensemble counterpart of the PDE formalism
needs to share weights. Table 1 below shows that, SWE also improves both natural and robust accuracies which verifies
the efficacy of our PDE formalism. Moreover, to further improve the ensemble model's performance, we generalize
SWE to non-share weights ensemble (NSWE) with the consideration of increasing the model capacity. As shown in
Table 1 in our paper ($En_2$ResNet20 v.s. ResNet44), NSWE remarkably outperforms the vanilla ResNets with a similar
capacity. We will point out this and include the results of SWE in the revision.

Table 1: Accuracy of the robustly trained $n\times En_1$ResNet20 which denotes the ensemble of $n$ share-weights $En_1$ResNet20.

|  | ResNet20 | $1\times$ $En_1$ResNet20 | $2\times$ $En_1$ResNet20 | $5\times$ $En_1$ResNet20 |
|---|---|---|---|---|
| $\mathcal{A}_{nat}$ | 75.11% | 77.21% | 77.88% | 77.99% |
| $\mathcal{A}_{rob}$ (IFGSM[20]) | 46.03% | 49.06% | 49.17% | 49.20% |

**Reviewer 3:** Do the authors run the experiments for multiple times observe consistent gain over the baseline?
**Reply:** The reported accuracies are averaged over five runs, and the standard deviation is less than 0.5% among these
runs. The accuracy of EnResNets is consistently better than the baseline over different runs.
**Reviewers 3 & 4:** Influence of $\epsilon$ for adversarial perturbation, and the standard deviation of the Gaussian noise.
**Reply:** We take the most-used $\epsilon$, and different $\epsilon$/noise will be discussed in the revision. Table 2 lists $\mathcal{A}_{rob}$ (IFGSM[20])
of the robustly trained $En_2$ResNet20 with different noise. 0.1 gives a good trade-off between accuracy and variance.

Table 2: Robust accuracy of the PGD adversarially trained $En_2$ResNet20 with different Gaussian noise injection. (five runs)

| Standard derivation (Gaussian noise) | 0.05 | 0.1 | 0.4 | 0.8 |
|---|---|---|---|---|
| $\mathcal{A}_{rob}$ (IFGSM[20]) | $50.05\% \pm 0.27\%$ | $50.06\% \pm 0.35\%$ | $50.51\% \pm 0.90\%$ | $43.51\% \pm 3.78\%$ |

**Reviewers 3 & 4:** 1). Specify the details of ResNets. 2). Reorganize Sec.3.3. 3). Provide derivation of Eq. (10). 4).
Line 126, better symbols for noise injected ResNet might be Resnet* etc. 5). Add the meanings of the symbols in Th 1.
**Reply:** We will modify our manuscript to include all these valuable suggestions.
**Reviewer 4:** The highest vanilla resnet accuracy in the paper is close to 85%, however, there are many implementations
of resnets which achieve $\sim 95\%$ on CIFAR10. Why aren't the baseline numbers reported in the paper high enough?
**Reply:** The reported accuracies are that of the robustly trained models by solving the EARM (Eq.(1)). The proposed
ResNets ensemble can also improve the natural accuracy of the naturally trained models, e.g., $\mathcal{A}_{nat}$ of naturally trained
$En_2$ResNet20 is 92.60% which is remarkably better than that of ResNet20 reported in He et al., arXiv:1512.03385.

[Meta-Review · NeurIPS 2019]

Good paper, accept. Please add the following points included in the rebuttal to the final version: - inference time and memory consumption - additional results on the robustness of the proposal with natural training. - additional results about the robustness of the methods.